# dCas9 regulator to neutralize competition in CRISPRi circuits

Hsin-Ho Huang [1,5], Massimo Bellato [2,5], Yili Qian [1], Pablo Cárdenas [3], Lorenzo Pasotti[2], Paolo Magni[2] & Domitilla Del Vecchio [1,4 ✉]

CRISPRi-mediated gene regulation allows simultaneous control of many genes. However, highly specific sgRNA-promoter binding is, alone, insufficient to achieve independent transcriptional regulation of multiple targets. Indeed, due to competition for dCas9, the repression ability of one sgRNA changes significantly when another sgRNA becomes expressed. To solve this problem and decouple sgRNA-mediated regulatory paths, we create a dCas9 concentration regulator that implements negative feedback on dCas9 level. This allows any sgRNA to maintain an approximately constant dose-response curve, independent of other sgRNAs. We demonstrate the regulator performance on both single-stage and layered CRISPRi-based genetic circuits, zeroing competition effects of up to 15-fold changes in circuit I/O response encountered without the dCas9 regulator. The dCas9 regulator decouples sgRNA-mediated regulatory paths, enabling concurrent and independent regulation of multiple genes. This allows predictable composition of CRISPRi-based genetic modules, which is essential in the design of larger scale synthetic genetic circuits.

[1] Department of Mechanical Engineering, Massachusetts Institute of Technology, Cambridge, MA, USA. [2] Laboratory of Bioinformatics, Mathematical Modelling and Synthetic Biology, Department of Electrical, Computer and Biomedical Engineering, University of Pavia, Pavia, Italy. [3] Department of Biological Engineering, Massachusetts Institute of Technology, Cambridge, MA, USA. [4] Synthetic Biology Center, Massachusetts Institute of Technology, Cambridge, MA, USA. [5] These authors contributed equally: Hsin-Ho Huang, Massimo Bellato. ✉email: ddv@mit.edu

The clustered regularly interspaced short palindromic repeats interference (CRISPRi) allows to create many orthogonal transcriptional repressors, each constituted of catalytically inactive Cas9 (dCas9) bound to a single guide RNA (sgRNA) ($c_i$ in Fig. 1a). This is instrumental to modulate complex transcriptional programs[1,2] and to build larger scale synthetic genetic circuits[3,4]. Multiplexing is possible because many sgRNAs can be concurrently expressed to recruit dCas9 each to a different promoter with high specificity to regulate transcription[5]. High specificity of sgRNA-promoter binding allows the creation of a large library of transcriptional repressors that, in principle, do not interfere with one another[6]. High specificity combined with lower loading to gene expression resources than with protein-based transcription factors[7] makes the CRISPRi-dCas9 system an optimal solution to create increasingly large and sophisticated transcriptional programs[8].

Despite the high specificity of sgRNA-promoter binding, multiple sgRNAs can still interfere with one another by competing for binding to dCas9[9–12]. This binding imparts a load to dCas9, which affects the level of free dCas9 (apo-dCas9) (Fig. 1a). Specifically, Zhang et al. showed that the fold-repression exerted by any one sgRNA through CRISPRi can decrease by up to five times when additional sgRNAs are expressed[9]. In ref. [10], the authors observed that the ability of one sgRNA to repress its target was hampered when a second sgRNA without a target was constitutively expressed, consistent with predictions from mathematical models[12]. In these works, and more generally in all CRISPRi-dCas9 transcriptional programs to date, dCas9 protein is generated at a constant pre-fixed rate. As a consequence, the concentration of dCas9 bound to any sgRNA species decreases as additional sgRNAs that bind to dCas9 become expressed. This leads to undesirable coupling among theoretically orthogonal sgRNA-mediated regulatory paths, wherein each sgRNA modulates directly the transcription of its targets but also indirectly affects transcription of non-target genes[12]. This interference confounds design since the combined input/output (I/O) response of multiple CRISPRi modules (CM) operating concurrently in the cell is different from that predicted using the I/O responses of each CM characterized in isolation. Although the effects of dCas9 loads may, in principle, be mitigated by increased levels of dCas9 (see simulations in Supplementary Fig. 1), it is difficult to increase dCas9 level in practice because this causes severe growth defects (see refs. [6,13,14] and Supplementary Note 2). To address this problem, less toxic mutations of dCas9 protein have appeared. Yet, even at the allowed higher dCas9 levels, the effects of dCas9 loading remain prominent[9].

In this work, we present a regulated dCas9 generator that adjusts the production rate of dCas9 such that the concentration

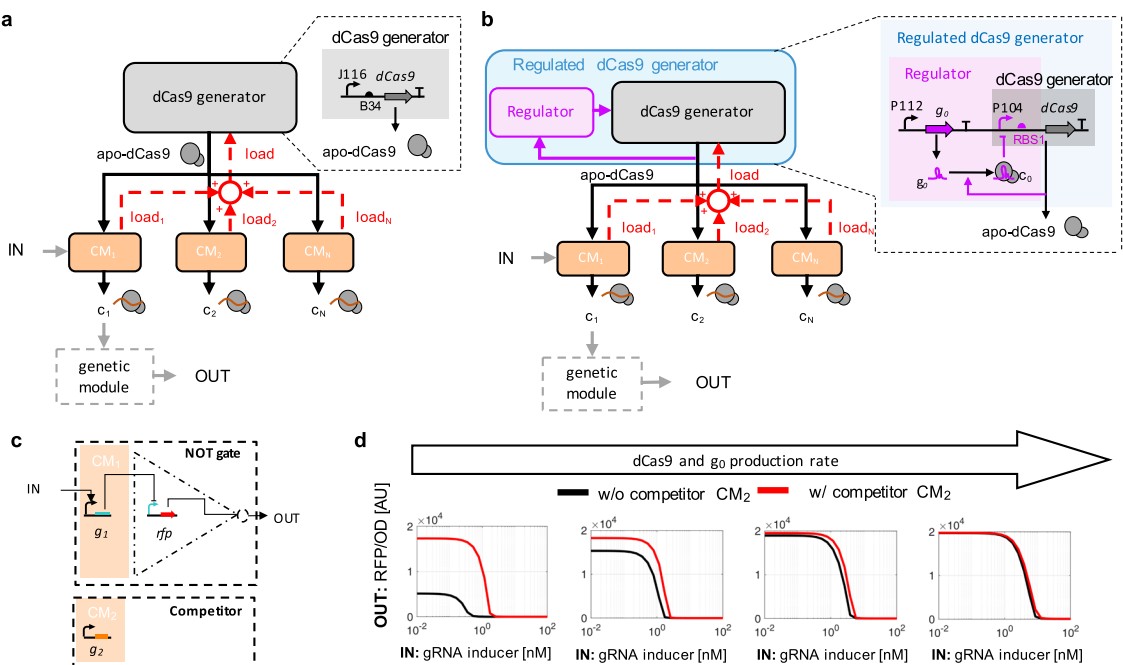

**Fig. 1 Regulated dCas9 generator mitigates loads in CRISPRi-based genetic circuits. a** Block diagram of unregulated (UR) dCas9 generator subject to load and corresponding genetic implementation. An unregulated dCas9 generator produces apo-dCas9 at a pre-fixed constant rate. A CRISPRi module (CM) takes apo-dCas9 as a resource input, comprises a sequence specific sgRNA, and produces the sgRNA-dCas9 complex $c_i$ as a transcriptional repressor output. A CM typically takes regulatory inputs (see $CM_1$) that control the expression of the sgRNA. The repressor $c_i$ can serve as a regulatory input to another CM or to a module expressing a protein (genetic module). The sequestration of dCas9 by each sgRNA is graphically represented as a load that the corresponding CM applies to the dCas9 generator (red dashed arrows). The overall load to the dCas9 generator is the sum (open red circle) of all the loads by each CM. In the genetic implementation, the dCas9 gene is expressed by the BBa_J23116 promoter (shown as J116) and the RBS BBa_B0034 (shown as B34). **b** Block diagram of regulated (R) dCas9 generator subject to load and corresponding genetic implementation. A regulated dCas9 generator produces apo-dCas9 at a rate that is adjusted based on the level of apo-dCas9 itself via a regulator. In the genetic implementation, the regulator comprises constitutive expression (through promoter P112) of gRNA $g_0$, which binds to dCas9 to form the dCas9-$g_0$ complex $c_0$. This, in turn, represses dCas9 promoter (P104) through CRISPRi. dCas9 promoter P104 and RBS (RBS1) are both stronger than those of the unregulated generator. Details about parts and plasmids are reported in Supplementary Notes 3 and 6. **c** Logic diagram of a CRISPRi-based NOT gate. The logic is implemented through the repression of a constitutive promoter driving a reporter gene ($rfp$) by an inducible sgRNA. **d** Parameters controlling the performance of the regulated dCas9 generator. Simulation results show the I/O response of a CM-based NOT gate in the presence or absence of the competitor CM (see **c**), as the production rate of $g_0$ and of dCas9 are increased. The left-side panel corresponds to a regulated dCas9 generator with no $g_0$ making the system like an unregulated dCas9 generator. Details about mathematical model and parameters are reported in Supplementary Note 1.

of apo-dCas9, and thus that of dCas9 bound to any sgRNA species, becomes practically independent of the load (Fig. 1b). This, in turn, ensures an approximately constant repression strength of any sgRNA when additional sgRNAs become expressed, thus decoupling regulatory paths of different sgRNAs. We create an experimental model system that recapitulates the dCas9 loading problem, in which dCas9 is produced by either an unregulated (UR) (Fig. 1a) or a regulated (R) (Fig. 1b) dCas9 generator. In the unregulated dCas9 generator, the production rate of dCas9 is constant (Fig. 1a). By contrast, the regulated dCas9 generator produces dCas9 at a rate that is negatively regulated by the level of apo-dCas9 itself through CRISPRi by means of sgRNA $g_0$ (Fig. 1b). The regulator's performance is evaluated on genetic circuits that show severe alterations of their I/O responses when a competitor sgRNAs is expressed.

## Results

**Decoupling NOT gates' I/O responses from competitor sgRNA expression.** We first evaluated the performance of the regulated dCas9 generator by comparing the extent by which the I/O response of a CM-based logic NOT gate (Fig. 1c) is affected by the expression of a competitor sgRNA. The CM-based NOT gate is constituted of a CM with sgRNA $g_1$ expressed by an inducible promoter and a genetic module expressing red fluorescent protein (RFP) as an output. The sgRNA $g_1$ represses RFP's transcription through CRISPRi. A second CM (competitor CM) contains sgRNA $g_2$ expressed by a constitutive promoter. This competitor CM plays the role of any additional CM that may become activated in a system, such as from other logic gates[3,15]. When competitor sgRNA $g_2$ is present and binds to dCas9, the level of apo-dCas9 drops. In the regulated dCas9 generator, this drop reduces the concentration of the dCas9-$g_0$ complex $c_0$, thereby de-repressing dCas9 transcription. As a result, the level of apo-dCas9 increases, thus balancing the initial drop due to the load exerted by competitor sgRNA $g_2$. This ultimately allows the concentration of dCas9 bound to sgRNA $g_1$ in the NOT gate to remain approximately constant when competitor $g_2$ becomes expressed.

We created an ordinary differential equation (ODE) model of the regulated dCas9 generator, the CM-based logic NOT gate, and the competitor CM (Supplementary Eqs. (33–35)). Simulations of this model show that the unrepressed production rate of dCas9 and the production rate of sgRNA $g_0$ are the two key design parameters. When these production rates are both sufficiently high, expression of competitor sgRNA $g_2$ no longer affects the NOT gate I/O response (simulations in Fig. 1d). In-depth mathematical analysis of the regulated dCas9 generator ODE model further shows that the sensitivity of apo-dCas9 level to the expression rate of the competitor sgRNA $g_2$ can be arbitrarily diminished by picking a sufficiently large expression rate for sgRNA $g_0$ (Supplementary Note 1.3). This sensitivity reduction, in turn, mitigates the effects of competitor sgRNA $g_2$'s expression on the NOT gate I/O response, although potentially affecting the ability of sgRNA $g_1$ to fully repress its target (simulations in Supplementary Fig. 1 and experimental data in Supplementary Fig. 5d, e). By also increasing dCas9 protein production rate, sgRNA $g_1$ can fully repress its target without hampering robustness of the NOT gate I/O response to competitor sgRNA $g_2$'s expression (simulations in Supplementary Fig. 1 and experimental data in Supplementary Fig. 5e, f)). Therefore, our regulated dCas9 generator employs strong promoter for sgRNA $g_0$ and also higher dCas9's promoter and RBS strengths compared to those of the unregulated generator (compare Fig. 1a (UR) with Fig. 1b (R)).

We implemented the CM-based NOT gate by using an HSL-inducible promoter controlling the expression of $g_1$. The competitor CM was created by expressing a competitor sgRNA $g_2$ through a library of constitutive promoters (Fig. 2a and Supplementary Note 6). The competitor sgRNA $g_2$ was designed to target a DNA sequence not present in the circuit nor in the bacterial genome (as predicted by Benchling's sgRNA design tool, see Supplementary Note 5) to avoid additional interactions that could confound analysis[16]. We experimentally evaluated the performance of the regulated dCas9 generator by first building CMs in high copy number plasmids (~84 copies, Supplementary Note 6) in order to create large dCas9 loads. In the unregulated generator configuration, the choice of dCas9's promoter and RBS ensures production of dCas9 at a level sufficient to enable complete repression of target promoters by sgRNAs without substantially affecting growth rate (see ref. [17] and Supplementary Note 6, Supplementary Fig. 9, Supplementary Note 7). With the unregulated dCas9 generator, expression of the competitor sgRNA causes changes of 16, 17, or 42 fold in the I/O response of the CM-based NOT gate depending on the input level (Fig. 2b). By contrast, with the regulated dCas9 generator no appreciable change is observed (Fig. 2c), indicating that the regulated dCas9 generator can attenuate large dCas9 loads.

We then investigated to what extent dCas9 competition still affects the function of logic gates when the CMs are built on low copy number plasmids. Thus, the NOT gate and competitor CMs were both assembled on pSB4C5 plasmid (giving ~5 copies, Supplementary Note 6). Interestingly, even when the CMs are on a low copy number plasmid, expressing competitor sgRNA $g_2$ still leads to appreciable changes in the I/O response of the NOT gate, leading to a 2-fold change in the NOT gate's output when the unregulated generator is used (Fig. 2d). These changes disappear when the regulated dCas9 generator is employed (Fig. 2e).

We also tested the load mitigation property of the regulated dCas9 generator in an alternative bacterial strain (Supplementary Note 4) and adopting a different design of the NOT gate, which uses an aTc-inducible promoter to control the expression of sgRNA $g_1$. In this case and with CMs in high copy number plasmids, expressing a competitor sgRNA results in up to 13-fold change in the NOT gate's I/O response with unregulated dCas9 generator, while it still gives inappreciable change when the regulated generator is used (Supplementary Fig. 8).

**Decoupling layered circuits' I/O responses from competitor sgRNA expression.** To demonstrate that the ability of the regulated dCas9 generator to mitigate the effects of dCas9 loading is not specific to a single CM-based NOT gate, we built a layered logic circuit constituted of two NOT gates arranged in a cascade (Fig. 3a, b). NOT gate cascades are a prototypical example of layered logic gates and are ubiquitous in circuits computing sophisticated logic functions[1,3,6,15,18,19]. In the cascade design, the LuxR/HSL input activates expression of sgRNA $g_1$ in $CM_1$, which, in turn, represses the expression of sgRNA $g_3$ in $CM_3$ through CRISPRi. The output of $CM_3$ then represses a promoter expressing the RFP output protein (Fig. 3b). As before, a competitor sgRNA $g_2$ was included or omitted from the system ($CM_2$). The two CMs constituting the cascade and the competitor CM are all on a high copy number plasmid (~84 copies). The I/O response of the cascade was measured with or without the competitor sgRNA $g_2$ and with either the unregulated or regulated dCas9 generators. The I/O response of the cascade shows approximately a 4-fold change at low induction levels when the competitor sgRNA is included and the unregulated dCas9 generator is used (Fig. 3c). By contrast, the cascade's I/O response shows no appreciable

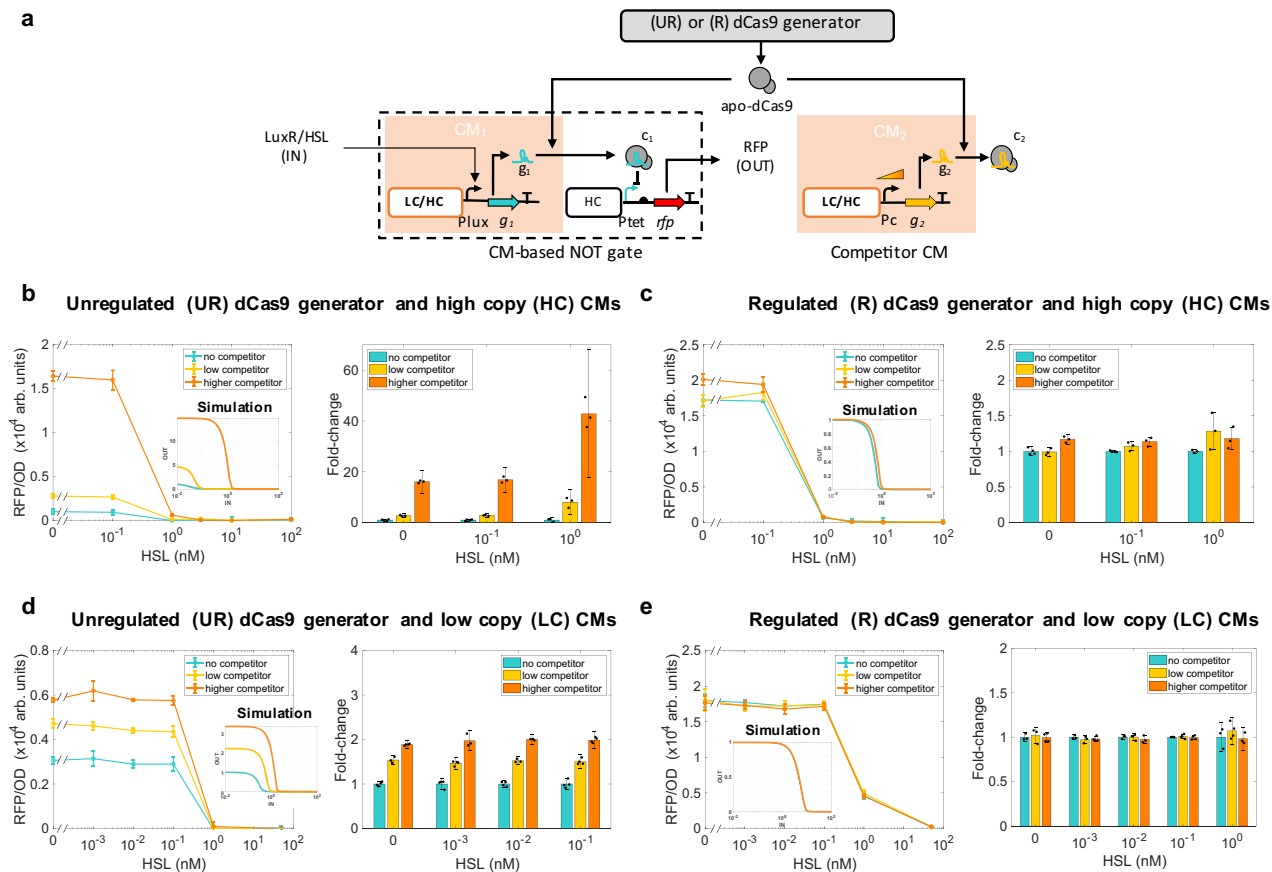

**Fig. 2 Effect of competitor CM on CM-based NOT gate with UR and R dCas9 generators. a** Genetic diagram of a CM-based NOT gate and competitor CM. The NOT gate is composed of a CM comprising sgRNA $g_1$, giving complex $c_1$ as an output, and of a genetic module that takes $c_1$ as input and gives RFP as an output. Here, $g_1$'s expression is regulated by HSL/LuxR (IN). A competitor CM that expresses competitor sgRNA $g_2$ with variable promoter Pc is either present or absent. The CMs were placed either on a low copy (LC) number plasmid (~5 copies) or on a high copy (HC) number plasmid (~84). The dCas9 generator was placed on a medium (~20) copy number plasmid (Supplementary Notes 6). **b** CM-based NOT gate I/O response with UR dCas9 generator and CMs on HC plasmid. Turquoise line represents I/O responses in the absence of competitor, while yellow (medium competitor) and orange (higher competitor) lines represent system I/O responses in which the expression of $g_2$ is driven by a weaker (BBa_J23100) or stronger (pTrc) promoter, respectively (Supplementary Note 6 and Supplementary Table 13). Inset shows steady state I/O responses obtained by simulating the ODE model of the unregulated system described by Supplementary Eqs. (30–32) with parameters in Supplementary Table 3. (**c**) CM-based NOT gate I/O response with R dCas9 generator and CMs on HC plasmid. Inset shows steady state I/O responses obtained by simulating the ODE model of the regulated system described in Supplementary Eqs. (33–35) with parameters in Supplementary Table 3. RFP distributions obtained by flow cytometry are reported in Supplementary Fig. 12. **d** CM-based NOT gate I/O response with UR dCas9 generator and CMs on LC plasmid. Yellow (medium competitor) and orange (higher competitor) lines represent system I/O responses in which the expression of competitor sgRNA is driven by a weaker (BBa_J23116) or stronger promoter (BBa_J23100), respectively (See Supplementary Note 6 and Supplementary Table 13). Inset shows steady state I/O responses obtained by simulating the ODE model of the unregulated system described by Supplementary Eqs. (30–32) with parameters in Supplementary Table 3. **e** CM-based NOT gate I/O response with R dCas9 generator and CMs on LC plasmid. Inset shows steady state I/O responses obtained by simulating the ODE model of the regulated system described in Supplementary Eqs. (33–35) with parameters in Supplementary Table 3. Data in the line plots represent mean values ± SD of $n = 3$ or 4 biologically independent experiments. Fold changes are normalized to the no competitor data. Bar graphs with error bars and overlaid dot plots are computed according to Eqs. (1–3) in "Methods" section. Insets' IN and OUT have the same units as in the data plots. Source data are provided with this paper.

change upon addition of the competitor when the regulated dCas9 generator is employed (Fig. 3d).

Therefore, the dCas9 regulator's ability to decouple the function of a circuit from expression of additional sgRNAs is independent of the specific genetic components of the circuit (i.e., promoters and inducers) and of whether the circuit contains one or multiple connected CMs, such as in layered systems.

## Discussion

In summary, our regulated dCas9 generator effectively decouples the regulatory functions of different sgRNAs from one another, despite that these sgRNAs compete for dCas9. This enables true

independent modulation of multiple genes simultaneously, which is not possible with commonly used unregulated dCas9 generators. Independent modulation of multiple genes by different sgRNAs is critical for scalability of CRISPRi-based genetic circuits. Indeed, scalability requires that the characteristics of any circuit stay unchanged when other circuits are added to the cell. With an unregulated dCas9 generator the I/O characteristics of various logic gates are severely affected when adding just one more CM to the cell (Figs. 2b, d and 3c). By contrast, with the regulated dCas9 generator, the gates' I/O characteristics are unperturbed by addition of the same CM (Figs. 2c, e and 3d). The regulated dCas9 generator is implemented in a dedicated plasmid (Supplementary Note 6 and Supplementary Fig. 9) and can

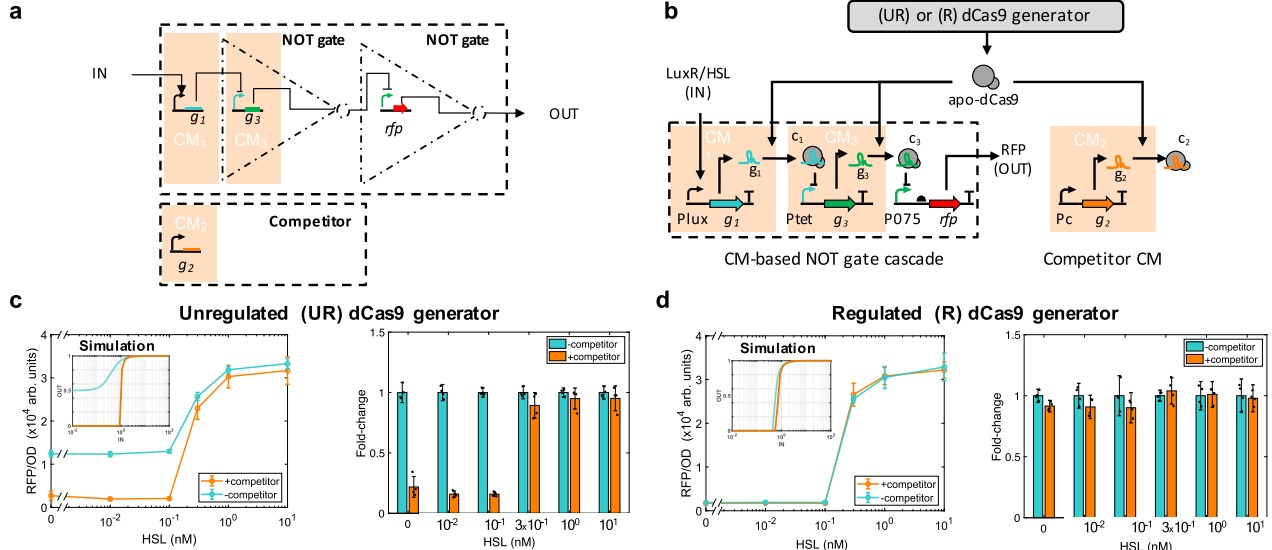

**Fig. 3 Effect of competitor CM on layered CM-based circuit with UR and R dCas9 generators. a** Logic diagram of a two-stage NOT gate cascade. Each NOT gate comprises repression of a constitutive promoter by an sgRNA. **b** Genetic circuit diagram of CM-based NOT gate cascade with either UR or R dCas9 generators. The cascade comprises $CM_1$ expressing sgRNA $g_1$, taking LuxR/HSL as input, and giving repressive complex $c_1$ as an output. Repressor $c_1$ then represses expression of sgRNA $g_3$ in $CM_3$, whose output $c_3$ represses RFP. The CMs are all on the same high copy number plasmid pSC101E93G (giving ~84 copies, Supplementary Note 6). The unregulated (UR) and regulated (R) dCas9 generators are as in Fig. 1a, b, respectively. Details about parts and plasmids are reported in Supplementary Note 6. **c** Effect of competitor CM on cascade's I/O response with UR dCas9 generator. Turquoise line represents cascade's I/O response in the absence of the competitor, while orange line represents I/O response in which the expression of competitor sgRNA is driven by a strong promoter (P108), (see Supplementary Note 6 and Supplementary Table 13). Inset shows steady state I/O responses obtained by simulating the ODEs listed in Supplementary Eqs. (36–38) with parameters given in Supplementary Table 3 where IN and OUT have the same units used in the data plots. Data in the line plots represent mean values ± SD of $n = 3$ or 4 biologically independent experiments. Fold changes are normalized to the no competitor data. Bar graphs with error bars and overlaid dot plots are computed according to Eqs. (1–3) in "Methods" section. **d** Effect of competitor CM on cascade's I/O response with R dCas9 generator. Color-coding of lines is as in **c**. Inset shows steady state I/O responses obtained by simulating the ODEs listed in Supplementary Eqs. (39–41) with parameters given in Supplementary Table 3. Insets' IN and OUT have the same units as in the data plots. Error bars are computed as in **c**. Source data are provided with this paper.

be easily transported across compatible bacterial strains and applications of CRISPRi-dCas9 systems.

The general problem of resource competition in genetic circuits has been widely studied in the context of competition for translation resources in bacteria[7,20,21] and for transcription resources in mammalian cells[22,23]. In particular, studies in bacteria have shown that the effects of ribosome competition on the I/O response of a genetic circuit can be very subtle, yet dramatic, as the inner circuit's modules compete with one another for ribosomes[20]. Similar phenomena are expected to also occur in more sophisticated CRISPRi-based genetic circuits[12]. Feedback regulation systems have been designed before to enhance robustness of bacterial genetic circuits to loading of gene expression resources, i.e., the ribosome[24–26]. However, none of these approaches is directly applicable to regulate apo-dCas9 level as they either use ribosome-specific parts[26], require the protein to be regulated (dCas9 in our case) to act as a transcriptional activator[24] or to sequester a transcriptional activator[25]. The dCas9 regulator is simple, compact, and exploits directly the ability of dCas9 to act as a transcriptional repressor when recruited to a promoter by sgRNAs. Feedback controllers enabled by CRISPRi have also appeared in other applications, where expression of a target gene is downregulated through a CRISPRi-mediated negative feedback to prevent growth rate defects due to over-expression[27]. When expressing multiple sgRNAs from the chromosome, i.e., in one copy, reduced loading effects are expected[28] and a regulated dCas9 generator may not be required in such cases. However, we have shown that for CRISPRi circuits constructed on plasmids, loading effects are prominent even at low plasmid copy number

(~5 copies, Fig. 2d). In these cases, it is expected that a regulated dCas9 generator will be required in order to ensure that multiple sgRNAs can concurrently and independently control their targets.

By combining high dCas9 expression rate with a strong feedback repression, our regulated dCas9 generator neutralizes any loading effect, while keeping dCas9 at sufficiently low levels to prevent growth defects (Supplementary Note 7). Nevertheless, transient toxicity is observed immediately after transformation of the regulated dCas9 generator plasmid into bacteria. In fact, immediately after this plasmid is transformed into cells, the initial concentrations of sgRNA $g_0$ and dCas9 are both zero, but dCas9 production rate is high and, due to a zero $g_0$ concentration, is initially unrepressed. Thus, dCas9 concentration increases rapidly after the plasmid is transformed and before $g_0$ level is sufficiently high to repress dCas9 transcription. This, in turn, may create an overshoot in dCas9 concentration resulting in toxic effects to cells. To decrease the overshoot in dCas9 concentration following transformation of the plasmid, we prepared host cells with a removable plasmid that produces the holocomplex dCas9-$g_0$. This represses dCas9 transcription from the plasmid encoding the regulated dCas9 generator immediately upon its transformation into cells, thus removing transient toxicity (see "Methods" section and Supplementary Note 2).

The adoption of multiple orthogonal dCas variants[29] could theoretically help mitigate the competition between two or more sgRNAs by distributing the resource demand among different DNA binding proteins. However, in large circuits composed of many CMs, it is still expected that multiple sgRNAs will need to share the same dCas variant. Indeed, since multiple variants are

expressed, it is also expected that the level of each single variant will be more limited, to prevent toxicity, than when using a single variant. Therefore, the effects of dCas competition can, in principle, be even more prominent than those observed here due to lower levels of the shared resource. In these situations, a dCas regulator will likely be required to neutralize competition. In general, a hybrid approach in which the variants shared among multiple sgRNAs include a regulator, and those that are not shared are not regulated could form an optimal solution. Although we did not investigate this aspect in this paper, the impact of resource competition on the dynamic behavior of a genetic circuit can also result in dramatic outcomes[30]. Further investigations are required to determine the benefit of the regulated dCas9 generator when dynamic behavior is of interest[31,32].

Overall, we expect that adoption of our regulated dCas9 generator in CRISPRi systems will make CRISPRi-based programs more scalable, reliable, and predictable.

## Methods

**Strain and growth medium**. Bacterial strain *E. coli* NEB10B (NEB, C3019I) was used in genetic circuit construction and characterization. The growth medium used in construction was LB broth Lennox. The growth medium used in characterization was M9 medium supplemented with 0.4% glucose, 0.2% casamino acids, and 1 mM thiamine hydrochloride. Appropriate antibiotics were added according to the selection marker of a genetic circuit. Final concentration of ampicillin, kanamycin, and chloramphenicol are 100, 25, and 12.5 $\mu g\,mL^{-1}$, respectively.

**Genetic circuit construction**. The genetic circuit construction was based on Gibson assembly method[33]. DNA fragments to be assembled were amplified by PCR using Phusion High-Fidelity PCR Master Mix with GC Buffer (NEB, M0532S), purified with gel electrophoresis and Zymoclean Gel DNA Recovery Kit (Zymo Research, D4002), quantified with the nanophotometer (Implen, P330), and assembled with Gibson assembly protocol using NEBuilder HiFi DNA Assembly Master Mix (NEB, E2621S). Assembled DNA was transformed into competent cells prepared by the CCMB80 buffer (TekNova, C3132). Plasmid DNA was prepared by the plasmid miniprep-classic kit (Zymo Research, D4015). DNA sequencing used Quintarabio DNA basic sequencing service. The lists of plasmids and primers are in Supplementary Table 7 and Supplementary Data, respectively. Due to the choice of a strong promoter and a strong ribosome binding site to express dCas9 in the regulated generator, toxicity transiently arises right after transformation before the $g_0$-dCas9 holocomplex has the time to form and repress dCas9's promoter. To prevent this transient toxicity, we first prepared competent cells that express $g_0$-dCas9 holocomplex from an auxiliary temperature-sensitive plasmid. The holocomplex targets and represses the dCas9 promoter on the regulated generator's plasmid right after transformation, thus preventing transient toxicity. Once the $g_0$-dCas9 holocomplex is expressed by the plasmid of the regulated dCas9 generator, the auxiliary plasmid is removed by plasmid curing at 42 °C (See Supplementary Note 2). The regulated dCas9 generator plasmid pdCas9_CL and the auxiliary temperature-sensitive plasmid pAUX_OL were deposited to Addgene as ID 166245 and 166246, respectively.

**Microplate photometer measurements**. Overnight culture was prepared by inoculating a −80 °C glycerol stock in 800 μL growth medium per well in a 24-well plate (Falcon, 351147) and grew at 30 °C, 220 rpm in a horizontal orbiting shaker for 13 h. Overnight culture was first diluted to an initial optical density at 600 nm ($OD_{600nm}$) of 0.001 in 200 μL growth medium per well in a 96-well plate (Falcon, 351172), and grew for 2 h to ensure exponential growth before induction. The 96-well plate was incubated at 30 °C in a Synergy MX (Biotek, Winooski, VT) microplate reader in static condition, and was shaken at a fast speed for 3 s right before OD and fluorescence measurements. Sampling interval was 5 min. Excitation and emission wavelengths to monitor RFP fluorescence were 584 and 619 nm, respectively. To ensure enough time to reach a steady state RFP/OD signal while the cells are in exponential growth, the cell culture was diluted with fresh growth medium to $OD_{600nm}$ of 0.01 when $OD_{600nm}$ approached 0.12 at the end of each batch. Multiple batches were conducted for a total experiment time of up to 25 h until RFP/OD reaches steady state (see Supplementary Note 9). The steady state I/O characteristics reported in the main text and the corresponding growth rates in Supplementary Note 7 were both computed from the last batch of each experiment.

**Quantification of competition effects**. To quantify competition effects, fold-change of a system's output at a given induced condition i was calculated by considering the ratio between the RFP/OD value of the system with competitor sgRNA ($g_2$) and that of the corresponding system bearing no competitor sgRNA (e.g., pOP94 and pCL112 for Fig. 2b, c). In particular, the height of the bars

represents mean fold-changes computed as follows:

$$\text{Mean fold}-\text{change (inducer = i)} = \frac{\sum_{j=1}^{N}[\text{RFP}_j/\text{OD}_j(\text{with } g_2, \text{inducer} = i)]/N}{\sum_{j=1}^{M}\left[\text{RFP}_j/\text{OD}_j(\text{without } g_2, \text{inducer} = i)\right]/M},$$

(1)

where subscript $j$ represents steady state RFP and OD measurements for the $j$-th biologically independent replicate; and $N$ and $M$ are the total number of replicates in the experiments with and without competitor $g_2$, respectively.

The size of the error bars represents one standard deviation (SD) of the respective fold-change. Specifically, for each induction level $i$, we account for propagation of uncertainty according to

$$\text{SD of fold}-\text{change} = \text{SD}\left(\frac{\text{RFP/OD with } g_2}{\text{RFP/OD without } g_2}\right)$$

$$= (\text{mean fold}-\text{change}) \cdot \sqrt{\text{CV}^2(\text{RFP/OD with } g_2) + \text{CV}^2(\text{RFP/OD without } g_2)},$$

(2)

where $\text{CV}^2(\text{RFP/OD with } g_2)$ and $\text{CV}^2(\text{RFP/OD without } g_2)$ are the coefficients of variation from biological replicates for steady state RFP/OD with and without competitor $g_2$, respectively.

The dot plots overlaid with bar graphs represent RFP/OD fold-changes for individual biological replicate with $g_2$ with respect to the mean of RFP/OD without $g_2$. Specifically, for the $j$th replicate at induction level $i$, we draw a dot according to:

$$\text{Replicate } j \text{ fold}-\text{change (inducer = i)} = \frac{\text{RFP}_j/\text{OD}_j(\text{with } g_2, \text{inducer} = i)]}{\sum_{j=1}^{M}\left[\text{RFP}_j/\text{OD}_j(\text{without } g_2, \text{inducer} = i)\right]/M}.$$

(3)

Note that by Eq. (2), the size of the error bars does not quantify the uncertainty in the dots defined according to Eq. (3), as the latter does not account for uncertainties in $\text{RFP}_j/\text{OD}_j(\text{without } g_2)$.

**Reporting summary**. Further information on research design is available in the Nature Research Reporting Summary linked to this article.

## Data availability

Simulation and fluorescence data generated or analyzed during this study are included in the paper and its Supplementary Information files. A reporting summary for this Article is available as a Supplementary Information file. The source data underlying Figs. 2 and 3 are provided as a Source Data file. Any other relevant data are available from the authors upon reasonable request.

## Code availability

Custom MATLAB (The MathWorks, Inc., Natick, MA, USA) codes are used to perform numerical simulations. A Supplementary Software file is provided, which includes codes to produce simulation results in Figs. 2 and 3.

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

## Acknowledgements

The authors want to thank Pin-Yi Chen (Massachusetts Institute of Technology, USA) for the modeling insights throughout the initial phase of the study and Michela Casanova (University of Pavia, Italy) for the assistance with preliminary experiments and cloning activities. This work was funded by NSF-CMMI Award # 1727189 and NSF-CCF Award # 2007674.

## Author contributions

H.H., M.B., and Y.Q. designed constructs and performed experiments. P.C. and H.H. devised the auxiliary pAUX_OL plasmid. Y.Q. and M.B. performed simulations and mathematical analyses. D.D.V., M.B., Y.Q., and H.H. wrote the paper and analyzed the data. P.M. and L.P. supervised the activities of M.B.; D.D.V. designed the research.

## Competing interests

The authors declare no competing interests.
