## [Peer Review File · Nature Communications]

Reviewers' Comments:

Reviewer #1:

Remarks to the Author:

The authors present a very nice piece of work on how to stabilize apo-dCas9 when there are multiple processes competing for the resource. The work is straightforward and results show the effectiveness of the negative feedback system used to stabilize the concentration of apo-dCas9. This is an important issue in the synthetic biology community.

I also liked very much how the modeling helped identify the dominant factors that contributed to the performance of the feedback system. To me, this was the most interesting part. It showed that biology can be engineered using normal engineering methods and one doesn't have to use trial and error to meet an objective.

Two minor points that should be addressed:

1. On page 2, third paragraph, the authors mention dCas9-g0 and complex c0, these symbols are not indicated in Figure 1d. I assume these symbols relate to that figure.
2. Given how important the modeling was to the paper I was extremely surprised to find that the authors did not provide source code for the model they used. I recommend they provide the scripts they used and the instructions on how to run them.

Reviewer #2:

Remarks to the Author:

The manuscript describes a strategy to reduce the effects of competition for a limited pool of dCas9 by multiple sgRNAs expressed simultaneously. This phenomenon was previously reported in Ref [6] and has been described by modelers in various papers. It is part of the broader challenge of operating biological processes under limited resources, which is important and has to be addressed in order to scale up synthetic biological circuitry.

This problem is illustrated here with an example system in which two sgRNA compete for dCas9: a reporter (RFP) is repressed via the CRISPRi method when sgRNA g1 is transcribed; when a second sgRNA g2 is produced, that competes for dCas9, the RFP dose response becomes altered, as shown via simulations (Fig.1) and experiments (Fig 2). To mitigate the adverse effects of competition for dCas9, the authors develop a strategy that relies on negative-autoregulation (feedback): dCas9 and an sgRNA responsible for dCas9 repression are both expressed at high rates, so that the average level of available dCas9 is less sensitive to fluctuations in demand of dCas9. The high-production, high-repression (feedback) strategy to mitigate competition for resources (perturbations) has been successfully used before by the authors, in the context of competition for ribosomes [18]. This idea is demonstrated here through experiments that compare the operation of CRISPRi constructs (repressors and a repressor cascade) in the presence of unregulated and regulated dCas9 generation, using plasmids at different copy numbers. The authors guide experimental results with through nicely derived mathematical models and computations. Appropriate controls are provided, and the data overall support the claims and results.

Here are my comments for improvement, roughly in order of importance.

1. The data in Fig. 2 clearly indicate that expression of the competitor sgRNA (g2) has an effect on the dose response of the sgRNA-driven repressor (g1). However, this effect seems peculiar to me because it occurs exclusively at low inducer levels causing a reduction of the "maximal" expression level of RFP, but no change in the repression threshold. Why should we care about loss of repression, when the repressor is only moderately induced? Since the repression threshold appears

unaffected, why don't we just calibrate the system to operate at high inducer levels? That will minimize the influence of competitors w/o need for additional circuitry. I do believe the shift in the low-inducer response matters and is important, especially when thinking of building interconnected dynamic circuits in which inducers may be endogenous species with fluctuating concentrations (a point that I actually do not think is mentioned). But because this study focuses on steady-state response, one wonders why this regulated dCas9 generator is necessary. The authors need to discuss this aspect.

2. Following up on comment 1, there is a noticeable discrepancy between the model prediction and the data: the model consistently predicts a shift in repression threshold in the presence of competitor for the unregulated system, but that shift is not observed in the data. Why is this the case? One possibility is that the QSS approximation made in the model does not hold for binding of gRNA and dCas9, and for binding of gRNA-dCas9 complex to the target promoter. If the authors are aware of published experimental data supporting the QSS assumption, they should cite it. I am aware of literature indicating rather that binding of gRNA-dCas9 is slow, and it can be as slow as 1 hour in the presence of other non-specific RNA molecules, see Mekler, Vladimir, et al. "Kinetics of the CRISPR-Cas9 effector complex assembly and the role of 3'-terminal segment of guide RNA." *Nucleic acids research* 44.6 (2016): 2837-2845.

3. The dangers of having dCas9 expressed at levels that are too high are mentioned in the discussion. There is also a note in the methods section about how the authors had to develop a two-step ad hoc protocol to successfully grow cells with the regulated dCas9 generator due to the high initial production of dCas9. I feel that this point should be made more prominent in the main text.

4. Page 2 lines 48–87 and Figure 1. The initial description of the problem, of the experimental system and of the modeling/computational results is organized in a way that I found a bit confusing, because a lot of technical/experimental details are provided for panels c and d, but then panel e shows simulations, and it seems all those details were a bit distracting relative to the main point of panel e. I suggest trying to start with the general challenge and the illustrative model/simulations, and later report the experimental details (Figs 1c and 1d could be greatly simplified, and their detailed version moved to Fig. 2).

5. I realize the goal of the paper is to propose a scalable strategy to manage competition for a single type of dCas9. Yet, I feel the authors should mention that there exist a large set of CRISPR-Cas variants that are in many cases orthogonal, and could be employed to mitigate competition in synthetic circuit subsystems. See, for example, Pickar-Oliver, Adrian, and Charles A. Gersbach. "The next generation of CRISPR-Cas technologies and applications." *Nature reviews Molecular cell biology* 20.8 (2019): 490-507.

6. In general, the list of references could be expanded to acknowledge additional recent literature toward developing methods for Cas9-based feedback & dynamic circuits. For example, a dCas9 feedback controller was characterized in Ceroni, Francesca, et al. "Burden-driven feedback control of gene expression." *Nature methods* 15.5 (2018): 387-393. This reference would complement archival ref [15] in regards to selection of appropriate levels of dCas9 expression to maintain a negligible burden. Dynamic circuits were recently demonstrated in Santos-Moreno, Javier, et al. "Multistable and dynamic CRISPRi-based synthetic circuits." *Nature Communications* 11.1 (2020): 1-8, where the authors actually speculate that competition may be beneficial for oscillator operation.

7. All figures have very poor resolution and especially insets have invisible details; this needs to be improved.

8. It is a good idea to be consistent with the x-axis of all figures and stick to the same range of HSL if possible; it sometimes changes between 0-10² (log scale, dose response) and 0-1 (linear, bar charts). I can see why one would want to focus on the most interesting range in each case, but this can lead to a collectively misleading representation of the results.

9. Fig. 1e – the units are missing from the axes. Units are also missing from other computational simulations in the insets of other figures.

10. SI p12, first lines – I think one obtains eq (6) by summing (5c) and (5e), not (5a) and (5c)

11. Fig. 1 caption (b) siagram->diagram

REVIEWER COMMENTS

Reviewer #1 (Remarks to the Author):

The authors present a very nice piece of work on how to stabilize apo-dCas9 when there are multiple processes competing for the resource. The work is straightforward and results show the effectiveness of the negative feedback system used to stabilize the concentration of apo-dCas9. This is an important issue in the synthetic biology community.

I also liked very much how the modeling helped identify the dominant factors that contributed to the performance of the feedback system. To me, this was the most interesting part. It showed that biology can be engineered using normal engineering methods and one doesn't have to use trial and error to meet an objective.

Two minor points that should be addressed:

1. On page 2, third paragraph, the authors mention dCas9-g₀ and complex c₀, these symbols are not indicated in Figure 1d. I assume these symbols relate to that figure.

We thank the reviewer to point out the missing symbol in the figure. New Panel b in Figure 1 now contains the symbol c₀ and the corresponding caption clarifies that it indicates the dCas9-g₀ complex.

2. Given how important the modeling was to the paper I was extremely surprised to find that the authors did not provide source code for the model they used. I recommend they provide the scripts they used and the instructions on how to run them.

Matlab source code implementing all the ODE models of the paper and a script that specifically runs the codes to produce the simulation plots appearing in the main text have been added as supplementary material.

Reviewer #2 (Remarks to the Author):

The manuscript describes a strategy to reduce the effects of competition for a limited pool of dCas9 by multiple sgRNAs expressed simultaneously. This phenomenon was previously reported in Ref [6] and has been described by modelers in various papers. It is part of the broader challenge of operating biological processes under limited resources, which is important and has to be addressed in order to scale up synthetic biological circuitry. This problem is illustrated here with an example system in which two sgRNA compete for dCas9: a reporter (RFP) is repressed via the CRISPRi method when sgRNA g₁ is transcribed; when a second sgRNA g₂ is produced, that competes for dCas9, the RFP dose response becomes altered, as shown via simulations (Fig.1) and experiments (Fig 2). To mitigate the adverse effects of competition for dCas9, the authors develop a strategy that relies on negative-autoregulation (feedback): dCas9 and an sgRNA responsible for dCas9 repression are both expressed at high rates, so that the average level of available dCas9 is less sensitive to fluctuations in demand of dCas9. The high-production, high-repression (feedback) strategy to mitigate competition for resources (perturbations) has been successfully used before by the authors, in the context of competition for ribosomes [18]. This idea is demonstrated here through experiments that compare the operation of CRISPRi constructs (repressors and a repressor cascade) in the presence of unregulated and regulated dCas9 generation, using plasmids at different copy numbers. The authors guide experimental results with through nicely derived mathematical models and computations. Appropriate controls are provided, and the data overall support the claims and results.

Here are my comments for improvement, roughly in order of importance.

1. The data in Fig. 2 clearly indicate that expression of the competitor sgRNA (g₂) has an effect on the dose

response of the sgRNA-driven repressor (g1). However, this effect seems peculiar to me because it occurs exclusively at low inducer levels causing a reduction of the “maximal” expression level of RFP, but no change in the repression threshold. Why should we care about loss of repression, when the repressor is only moderately induced? Since the repression threshold appears unaffected, why don’t we just calibrate the system to operate at high inducer levels? That will minimize the influence of competitors w/o need for additional circuitry. I do believe the shift in the low-inducer response matters and is important, especially when thinking of building interconnected dynamic circuits in which inducers may be endogenous species with fluctuating concentrations (a point that I actually do not think is mentioned). But because this study focuses on steady-state response, one wonders why this regulated dCas9 generator is necessary. The authors need to discuss this aspect.

We thank the reviewer for the insightful comments, highlighting aspects that should be clarified in the paper. Preventing loss of repression at *any* repression level, including at low repression, is critical in applications where tunability of the output is required instead of an ON/OFF response. For example, consider a system with two independently controlled sgRNAs meant to *modulate* (as opposed to switch ON or OFF – see [1]) two different targets independently. If induction of sgRNA₂ causes a loss of repression of sgRNA₁, this implies that there is a “hidden activation” from sgRNA₂ to the target of sgRNA₁ [2]. As a consequence, independent modulation of multiple targets by multiple inputs (multiplexing) is not possible.

More generally, preventing loss of repression at any repression level is crucial to ensure that multiple CRISPRi modules (CMs) can be composed together to obtain an outcome that can be predicted from the CMs input/output (I/O) responses characterized in isolation. In fact, if the I/O response of a CM changes when a second CM is added in the system, the combined I/O response of the two CMs (whether connected to each other or not) will be different from that predicted using the I/O responses characterized in isolation. This confounds system’s analysis and design and can be source of dramatic failure modes that are very difficult to dissect in complicated systems composed of multiple CMs. This issue is found in any genetic system where there is a shared limited resource, such as in genetic modules competing for a limited pool of ribosomes. Indeed, it was shown that, because of ribosome competition, the steady state I/O response of a two-stage genetic activation cascade could be monotonically decreasing or biphasic instead of monotonically increasing, as expected from composing the I/O responses of the two genetic modules [3].

To clarify these aspects, we reworded parts of the abstract and added the following sentence in the introduction (lines 49-52):

[...] , wherein each sgRNA modulates directly the transcription of its targets but also indirectly affects transcription of non-target genes [13]. This interference confounds design since the combined input/output (I/O) response of multiple CRISPRi modules (CM) operating concurrently in the cell is different from that predicted using the I/O responses of each CM characterized in isolation. [...]

A strategy where we adjust the inducer level for each sgRNA to obtain full output repression in the presence of competition is viable in applications where the sole objective is to shut down the expression of target genes. By contrast, this is not viable in applications where we want to independently modulate (as opposed to merely shut down) the expression of multiple genes concurrently, as described above. Furthermore, one major application of CRISPRi is to compose CMs together to create circuits with prescribed *I/O responses* (from the gRNA’s inducer(s) to the output protein(s) of interest), such as in NOR, AND, or NAND gates [4-9], as opposed to obtain a desired output. In these applications, the inducer level is not a free variable that can be used to obtain a desired output because our desired specification is the *I/O logic function*. Such a function, in turn, can be completely disrupted by dCas9 competition. In fact, as the reviewer highlights, dCas9 competition can hamper our ability to obtain predictable behavior of connected CMs and genetic modules. We remark that this disruptive effect of competition is encountered in *both dynamic and steady state behavior and not just in dynamic behavior*. As an example of the effect of competition on steady state I/O response, we recall the effects of ribosome competition in genetic activation cascades

[3]. Here, the I/O steady state response can be monotonically decreasing or biphasic instead of being monotonically increasing as expected from the I/O responses of the composing genetic modules. These failure modes pertaining to the steady state I/O response are present in CRISPRi circuits as well.

[Redacted]

Although we did not investigate this aspect in this paper, the impact of resource competition on the *dynamic behavior* of a genetic circuit, can result in even more dramatic outcomes as the reviewer points out. As an example, we refer the reviewer to a former paper of ours where we show that bistability of a genetic toggle switch can be destroyed by competition for ribosomes between the two nodes of the toggle switch [10]. The paper also shows that different topologies that realize bistability can be differentially affected by resource competition. To address the reviewer's comments about dynamic behavior being affected by resource competition, we have added the following sentence in the discussion at lines 184-187:

[...] Although we did not investigate this aspect in this paper, the impact of resource competition on the dynamic

behavior of a genetic circuit can also result in dramatic outcomes [17]. Further investigations are required to determine the benefit of the regulated dCas9 generator when dynamic behavior is of interest [32, 33]. [...]

2. Following up on comment 1, there is a noticeable discrepancy between the model prediction and the data: the model consistently predicts a shift in repression threshold in the presence of competitor for the unregulated system, but that shift is not observed in the data. Why is this the case? One possibility is that the QSS approximation made in the model does not hold for binding of gRNA and dCas9, and for binding of gRNA-dCas9 complex to the target promoter. If the authors are aware of published experimental data supporting the QSS assumption, they should cite it. I am aware of literature indicating rather that binding of gRNA-dCas9 is slow, and it can be as slow as 1 hour in the presence of other non-specific RNA molecules, see Mekler, Vladimir, et al. "Kinetics of the CRISPR-Cas9 effector complex assembly and the role of 3'-terminal segment of guide RNA." Nucleic acids research 44.6 (2016): 2837-2845.

We are grateful for the reviewer observation, which prompted us to acquire more data around the values of HSL where the target is almost fully repressed. In fact, we note that the repression threshold seemed unchanged by the competitor in the data in part because of the coarse HSL induction ladder, which skipped over the range where the difference could be observed. We picked a coarser induction ladder of HSL because in this paper we are focused on the performance of the dCas9 controller and not as much on the quantitative characterization of competition effects. Nevertheless, to address the reviewer's comments, we performed additional experiments for both the low copy and high copy NOT gates using a finer ladder of HSL around the value showing full repression. This data is shown in Figure R2, also reported in Supplementary Note 8:

Figure R2. Data for the NOT gate I/O steady state response for HSL values within the interval [0.1nM,1nM]. The NOT gate is the same as that of Figure 2a in the main text. The dashed horizontal line corresponds to 10% of the RFP/OD value obtained for 0nM HSL in the ‘no competitor’ condition.

Due to the stiffness of the CRISPRi NOT gate I/O response curve, our former HSL level fell around the “knees” of the curves, i.e., at HSL=0.1nM, just before the drop of the earliest curve – no competitor – and at HSL= 1nM, just above the full repression threshold of the latest curve – higher competitor. We therefore collected datapoints within this interval by adding 4 inductions at HSL = 0.2nM, 0.3nM, 0.4nM, and 0.6nM. To assess whether the “repression threshold” increases with competitor, we defined the *repression threshold (RT)* as the minimum HSL level required to obtain an RFP/OD output that is at or below 10% of the RFP/OD value obtained for HSL = 0nM in the ‘no competitor’ condition. Referring to Figure R2, we obtain the following estimates for the RT in the high-copy NOT

gate: $\sim 0.2\text{nM}$ for no competitor and low competitor, and greater than 1nM for high competitor. For the low copy NOT gate: $\sim 0.2\text{nM}$ for no competitor, and $\sim 0.3\text{nM}$ for low and high competitor. Hence, the RT increases with the addition of competitor gRNA.

Regarding the comment about the QSS assumption, all the data acquisition was conducted to ensure that the system had reached steady state of RFP/OD. Specifically, data were collected for multiple batches by diluting back the culture at the end of each batch to ensure exponential growth for up to a total experiment duration of 25 hours (see ONLINE METHODS). This ensured that the RFP/OD signal reached a constant (steady) level with time (see Supplementary Note 9). Because our data describes the steady state of the system, all our reported simulation results in the main text and SI, as well as the model guided design of the regulator (Supplementary Note 1.3), consider the ODE model's steady state, that is, when $d/dt=0$ for all of the species concentrations. As a consequence, any quasi-steady state (QSS) assumption made is not affecting the reported simulation results since $d/dt=0$ for all of the species concentrations and not just for those of the fast variables.

The values of the K_{on} and K_{off} of the rate constants of the binding and unbinding, respectively, between dCas9 and sgRNA affect the steady state of the system only through the constant $K=(K_{off}+\delta)/K_{on}$ of the binding (see Supplementary Note 1). According to the *in vitro* data in Figure 5 of the paper suggested by the reviewer, the 'effective' K_{on} can decrease substantially when competing RNA molecules are present. This, in turn, causes an increase of the value of the constant K . To address the reviewer's comment, we therefore investigated how increased values of K impact the effects of competition and the ability of the dCas9 generator to attenuate such effects. This is shown in Figure R3 (for the unregulated dCa9 generator) and in Figure R4 (for the regulated dCas9 generator), in which we increased K by up to two orders of magnitude with respect to the value of the paper (0.01nM):

Figure R3. NOT gate I/O steady state characteristic with unregulated dCas9 generator for different values of the dissociation constants between gRNA and dCas9 (K) and of the dissociation constants between gRNA:dCas9 complex with DNA (Q). The presence of competitor gRNA leads to slightly different dose response curves of the NOT gate when K is changed. All parameters except for K and Q are identical to those for the high copy NOT gate reported in Supplementary Table 3 and Supplementary Table 12.

Figure R4. NOT gate I/O steady state characteristic with regulated dCas9 generator for different values of the dissociation constants between gRNA and dCas9 (K) and of the dissociation constants between gRNA:dCas9 complex with DNA (Q). Robustness of the NOT gate to the presence of competitor gRNA is essentially independent of dissociation constants K and Q . All parameters except K and Q are identical to those for the high copy NOT gate reported in Supplementary Table 3 and Supplementary Table 12.

These figures show that the qualitative effects of competition and the ability of the regulated dCas9 generator to mitigate them are essentially independent of the values of K even if highly variable. We included in Supplementary note 1.4 these figures, along with the reference suggested by the reviewer.

3. The dangers of having dCas9 expressed at levels that are too high are mentioned in the discussion. There is also a note in the methods section about how the authors had to develop a two-step ad hoc protocol to successfully grow cells with the regulated dCas9 generator due to the high initial production of dCas9. I feel that this point should be made more prominent in the main text.

We thank the reviewer for this observation. In order to highlight the care that should be taken when transforming the dCas9 generator into cells to avoid toxicity, we expanded the description in the Methods and included also a comment on this point in the discussion at lines 166-175:

[...] Nevertheless, transient toxicity is observed immediately after transformation of the regulated dCas9 generator plasmid into bacteria. In fact, immediately after this plasmid is transformed into cells, the initial concentrations of sgRNA g_0 and dCas9 are both zero, but dCas9 production rate is high and, due to a zero g_0 concentration, is initially unrepressed. Thus, dCas9 concentration increases rapidly after the plasmid is transformed and before g_0 level is sufficiently high to repress dCas9 transcription. This, in turn, may create an overshoot in dCas9 concentration resulting in toxic effects to cells. To decrease the overshoot in dCas9 concentration following transformation of the plasmid, we prepared host cells with a removable plasmid that produces the holocomplex dCas9- g_0 . This represses dCas9 transcription from the plasmid encoding

the regulated dCas9 generator immediately upon its transformation into cells, thus removing transient toxicity (see ONLINE METHODS and Supplementary Note 2). [...]

4. Page 2 lines 48—87 and Figure 1. The initial description of the problem, of the experimental system and of the modeling/computational results is organized in a way that I found a bit confusing, because a lot of technical/experimental details are provided for panels c and d, but then panel e shows simulations, and it seems all those details were a bit distracting relative to the main point of panel e. I suggest trying to start with the general challenge and the illustrative model/simulations, and later report the experimental details (Figs 1c and 1d could be greatly simplified, and their detailed version moved to Fig. 2).

We thank the reviewer for this suggestion. We followed it and re-arranged Figure 1, Figure 2, and the main text pertaining to them. Specifically, we removed the details of the genetic implementation of the NOT gate from Figure 1 and moved it in Figure 2. In Figure 1, we only kept a high-level representation of the CM-based NOT gate and the genetic implementation of the regulated dCas9 generator. This is useful in this picture as it is referred to in panel d to pinpoint the parameters being changed in the simulation for model-guided design.

5. I realize the goal of the paper is to propose a scalable strategy to manage competition for a single type of dCas9. Yet, I feel the authors should mention that there exist a large set of CRISPR-Cas variants that are in many cases orthogonal, and could be employed to mitigate competition in synthetic circuit subsystems. See, for example, Pickar-Oliver, Adrian, and Charles A. Gersbach. "The next generation of CRISPR-Cas technologies and applications." *Nature reviews Molecular cell biology* 20.8 (2019): 490-507.

We thank the reviewer for providing this suggestion and the associated reference. In theory, if we were able to use a different dCas variant for each gRNA and each dCas9 variant were only binding its supposed sgRNA and none of the others, then we would remove competition. In practice, dCas-gRNA binding will likely not be fully orthogonal. Additionally, for creating larger circuits composed of many sgRNAs, it may still be required that one dCas variant is used for multiple sgRNAs. In this situation, competition among these sgRNAs will still arise. This competition may be even more severe than what observed now since the level of each dCas variant will most likely be limited to values not exceeding those of dCas9 in current applications. In fact, toxicity of each dCas variant will sum up over all dCas variants used such that each dCas expression will need to be limited to levels lower than those reachable if only one variant were used. That said, these are all very interesting tradeoffs to explore and a combined approach of using multiple variants along with a dCas regulator for some of them may be optimal. We added a few sentences about this in the discussion in lines 176-184:

[...] The adoption of multiple orthogonal dCas variants [31] could theoretically help mitigate the competition between two or more gRNA by distributing the resource demand among different DNA binding proteins. However, in large circuits composed of many CMs, it is still expected that multiple sgRNAs will need to share the same dCas variant. Indeed, since multiple variants are expressed, it is also expected that the level of each single variant will be more limited, to prevent toxicity, than when using a single variant. Therefore, the effects of dCas competition can, in principle, be even more prominent than those observed here due to lower levels of the shared resource. In these situations, a dCas regulator will likely be required to neutralize competition. In general, a hybrid approach in which the variants shared among multiple sgRNAs include a regulator and those that are not shared are not regulated could form an optimal solution. [...]

6. In general, the list of references could be expanded to acknowledge additional recent literature toward developing methods for Cas9-based feedback & dynamic circuits. For example, a dCas9 feedback controller was characterized in Ceroni, Francesca, et al. "Burden-driven feedback control of gene expression." *Nature methods* 15.5 (2018): 387-393. This reference would complement archival ref [15] in regards to selection of appropriate levels of dCas9 expression to maintain a negligible burden. Dynamic circuits were recently demonstrated in

Santos-Moreno, Javier, et al. "Multistable and dynamic CRISPRi-based synthetic circuits." *Nature Communications* 11.1 (2020): 1-8, where the authors actually speculate that competition may be beneficial for oscillator operation.

We thank the reviewer for these suggestions. We have broadened the cited literature to studies of resource competition, burden, and included the references suggested by the reviewer in the discussion. Some sentences on resource sharing have been added and rephrased in the introduction as well with proper references.

In particular, the two above references were cited as follows in lines 145-150 and 184-187, respectively:

[...] The general problem of resource competition in genetic circuits has been widely studied in the context of competition for translation resources in bacteria [10, 16] and for transcription resources in mammalian cells [18, 19]. In particular, studies in bacteria have shown that the effects of ribosome competition on the I/O response of a genetic circuit can be very subtle, yet dramatic, as the inner circuit's modules compete with one another for ribosomes. [16]. Similar phenomena are expected to also occur in more sophisticated CRISPRi-based genetic circuits [13]. [...]

[...] Feedback controllers enabled by CRISPRi have also appeared in other applications where expression of a target gene is down-regulated through a CRISPRi-mediated negative feedback to prevent growth rate defects due to over-expression [12]. [...]

7. All figures have very poor resolution and especially insets have invisible details; this needs to be improved.

We noticed that the resolution was poor due to an error in the conversion of the figures to a low-quality format. Figures have now been uploaded as “.pdf/.eps” files, with higher resolution.

8. It is a good idea to be consistent with the x-axis of all figures and stick to the same range of HSL if possible; it sometimes changes between 0-10² (log scale, dose response) and 0-1 (linear, bar charts). I can see why one would want to focus on the most interesting range in each case, but this can lead to a collectively misleading representation of the results.

We agree with the reviewer and parsed the plots to fix consistency. We just want to point out that the displayed bars in the bar charts were selected to report only fold changes that could be computed with reasonable error. In fact, when RFP/OD reaches values close to zero for HSL greater than 1nM the fold change, due to division by a very small number, would show a very large error bar.

9. Fig. 1e – the units are missing from the axes. Units are also missing from other computational simulations in the insets of other figures.

Units in figure 1e have been added. Units in the insets have been omitted to avoid figure overcrowding. However, figure captions have been changed adding the sentence:

[...] where IN and OUT have the same units used in the data plots [...]

10. SI p12, first lines – I think one obtains eq (6) by summing (5c) and (5e), not (5a) and (5c)

Thanks for pointing it out, the reference has been fixed.

11. Fig. 1 caption (b) siagram->diagram

We apologize for the typo which has been fixed.

BIBLIOGRAPHY

- [1] Fontana, J., Dong, C., Ham, J.Y., et al. Regulated Expression of sgRNAs Tunes CRISPRi in *E. coli*. *Biotechnol J* 13(9), 2020.
<https://doi.org/10.1002/biot.201800069>
- [2] Chen, P.Y., Qian, Y. & Del Vecchio, D. A model for resource competition in CRISPR-mediated gene repression. *BioRxiv*, 2018.
<https://doi.org/10.1101/266015>
- [3] Qian, Y., Huang, H.H., Jiménez, J. et al. Resource Competition Shapes the Response of Genetic Circuits. *ACS Synth Biol* 6(7), 2017. <https://doi.org/10.1021/acssynbio.6b00361>
- [4] Nielsen, A.A.K. and Voigt, C.A. Multi-input CRISPR/Cas genetic circuits that interface host regulatory networks. *Molecular Systems Biology*, 10(11), 2014.
 - a. <https://doi.org/10.15252/msb.20145735>
- [5] Gander, M.W., Vrana, J.D., Voje, W.E. et al. Digital logic circuits in yeast with CRISPR-dCas9 NOR gates. *Nature Communications*, 8(1), 2017
 - a. <https://doi.org/10.1038/ncomms15459>
- [6] Didovyk, A., Borek, B., Hasty, J. and Tsimring, L. Orthogonal modular gene repression in *Escherichia coli* using engineered CRISPR/cas9. *ACS Synthetic Biology*, 5(1), 2015.
 - a. <https://doi.org/10.1021/acssynbio.5b00147>
- [7] Gao, Y., Xiong, X., Wong, S., et al. Complex transcriptional modulation with orthogonal and inducible dCas9 regulators. *Nature Methods*, 13(12), 2016.
 - a. <https://doi.org/10.1038/nmeth.4042>
- [8] Kiani, S., Beal, J., Ebrahimkhani, M.R. et al. CRISPR transcriptional repression devices and layered circuits in mammalian cells. *Nature Methods*, 11(7), 2014. <https://doi.org/10.1038/nmeth.2969>
- [9] Weinberg, B.H., Hang Pham, N.T., Caraballo, L.D., et al. Large-scale design of robust genetic circuits with multiple inputs and outputs for mammalian cells. *Nature Biotechnology*, 35(5), 2017.
<https://doi.org/10.1038/nbt.3805>
- [10] McBride, C., Del Vecchio, D. Trade-offs in Robustness to Perturbations of Bacterial Population Controllers. *BioRxiv*, 2020.
<https://doi.org/10.1101/2020.06.04.134932>

Reviewers' Comments:

Reviewer #2:

Remarks to the Author:

The revised manuscript addresses my comments very clearly. One last suggestion is to modify the paragraph at lines 67-82 to begin by stating that models were used as a first tool to compare the R and UR system (Fig 1e). Right now this is only said at the end of the paragraph.

Response to reviewers

Reviewer #2 (Remarks to the Author):

The revised manuscript addresses my comments very clearly. One last suggestion is to modify the paragraph at lines 67-82 to begin by stating that models were used as a first tool to compare the R and UR system (Fig 1e). Right now this is only said at the end of the paragraph.

Response: We would like to thank the reviewer for the suggestion. After careful evaluation, we have opted for leaving the beginning of that paragraph unchanged and to instead move to a second paragraph the description of the model/simulations. The reason is because the first paragraph is meant to introduce the dCas9 regulator and the method that we use to evaluate its performance, that is, by comparing open and closed loop systems data. This comparison is used for all the experimental results of the paper and, in particular, also for the modeling results as described at the end of the old paragraph (now the beginning of the new one). We believe that the new subdivision into two paragraphs clarifies the portions of the statements that pertain to the model/simulations.